# Physiological Responses of *Agave maximiliana* to Inoculation with Autochthonous and Allochthonous Arbuscular Mycorrhizal Fungi

**DOI:** 10.3390/plants12030535

**Published:** 2023-01-24

**Authors:** Laura Verónica Hernández-Cuevas, Luis Alberto Salinas-Escobar, Miguel Ángel Segura-Castruita, Paola Andrea Palmeros-Suárez, Juan Florencio Gómez-Leyva

**Affiliations:** 1Laboratorio de Biología Molecular, TecNM-Instituto Tecnológico de Tlajomulco, Km 10 Carretera a San Miguel Cuyutlán, Tlajomulco de Zúñiga 45640, Jalisco, Mexico; 2Licenciatura en Biología, Universidad Autónoma de Tlaxcala, Km 10.5 Carr. Texmelucan-Tlaxcala, Ixtacuixtla de Mariano Matamoros 90122, Tlaxcala, Mexico; 3Centro Universitario de Ciencias Biológicas y Agropecuarias, Universidad de Guadalajara, Zapopan 44600, Jalisco, Mexico

**Keywords:** AMF native consortium, *Claroideoglomus claroideum*, *Claroideoglomus etunicatum*, biofertilizer, *Agave maximiliana*

## Abstract

The benefits of mycorrhizal interactions are only known in 8 of 210 recognized *Agave* taxa. We evaluated the effects of autochthonous and allochthonous arbuscular mycorrhizal fungi (AMF) on growth and nutrient assimilation in *Agave maximiliana*. The autochthonous consortium (Cn) of eight species was propagated from the rhizospheric soil of *A. maximiliana*, while *Claroideoglomus claroideum* (Cc) and *Claroideoglomus etunicatum* (Ce) were employed as allochthonous AMF. Six treatments were included in the study: Cn, Ce, Cc, Ce + Cc, Tf (fertilized control), and Tn (non-fertilized control, not inoculated). Mycorrhizal colonization increased over time, and the colonization percentages produced by Cn and the allochthonous AMF, both alone and mixed together, were equal at 6, 12, and 18 months. Height increased steadily and was higher in AMF-treated plants from seven months onward. Growth indicators of AMF-treated and AMF-free plants were equal at 6 months, but the beneficial effects of allochthonous and autochthonous AMF were evident in all growth indicators at 18 months and in sugar and mineral (P, K, Ca, Mg, and Fe) content. Arbuscular mycorrhizal fungi significantly improved all growth parameters of *A. maximiliana* regardless of the origin of the inoculums. This is the first study to report the positive effects of AMF colonization in *A. maximiliana*.

## 1. Introduction

Mexico is known for its great diversity of *Agave* plants. Indeed, of the 210 *Agave* species that have been identified, approximately 75% (159 species) are present in Mexico. Moreover, 129 of these species (61% of all *Agave* species worldwide and 81% of those distributed within Mexico) are endemic to the country [1]. The *Agave* genus is not only taxonomically rich but also culturally rich. Part of the cultural richness of Agave plants comes from the fibers, textiles, syrups, soaps, juices, and alcoholic beverages that are created from different parts of the plants [2]. 

At the ripening stage, which is marked by the appearance of inflorescence Agave plants accumulate substantial amounts of carbohydrates (i.e., oligofructans) in their stems. After being extracted, hydrolyzed, fermented, and distilled, these oligofructans are used to produce various alcoholic beverages (e.g., bacanora, mezcal, tequila, and raicilla). Many of these alcoholic beverages are associated with their particular regions of origin. For example, raicilla originates from the western coast-sierra region of Jalisco. There, raicilla production is profitable due to local consumption, and the economic resources generated by raicilla sales complement those of local agricultural activities [3].

From 2021 to 2022, the export of raicilla grew by 307%, with the principal destination being the United States and other destinations including Germany, Australia, Spain, and Italy. During this time, raicilla exports went from bringing in 64,316 USD to 261,637 USD [4]. For the purposes of raicilla production, *Agave* plants must be 6 or 7 years old and mature (~80% of the raw material). Raicilla can be made from multiple *Agave* species incluiding *A. angustifolia*, *A. rhodacantha*, *A. valenciana*, *A. inaequidens*, and *A. maximiliana*, although the latter two species are the ones typically used to make this distilled beverage [5]. While many *Agave* species are currently being domesticated, much of the production of raicilla still stems from wild plants. The wild populations of raicilla species have recently been put under additional stress due to the newly awarded Designation of Origin of Raicilla [6]. This designation has increased the pressure on wild *Agave* plant populations to meet the demand for raicilla. To facilitate the extensive cultivation of these species as a means to mitigate the negative effects on wild plants, it is necessary to develop sustainable production systems that do not rely on extraction. Indeed, it is more important than ever to develop management strategies that will conserve the genetic resources of these important plants. 

Beneficial microorganisms can favor the domestication and cultivation of plant species by supporting fundamental biogeochemical processes that affect soil fertility and plant health and productivity [7]. Nitrogen-fixing bacteria, rhizobacteria that favor plant growth, and fungi that form arbuscular mycorrhiza are some key microbial groups that promote and support plant development [8]. Agave plants are known to host arbuscular mycorrhizal fungi (AMF), yet the influence of AMF on the growth and development of *Agave* species has not been adequately explored. In fact, AMF interactions have only been studied in only 8 of 159 *Agave* species that exist in Mexico.

*Agave deserti* was the first *Agave* species to be used to study the effects of inoculation with autochthonous AMF. Inoculation was found to increase water conductivity, CO_2_ assimilation, and the content of P and Zn content in the roots and stems of *A. deserti,* respectively [9]. When inoculated with a consortium of autochthonous AMF (i.e., a mixture of AMF from the rhizosphere of the species under study) or allochthonous AMF (*Rhizophagus intraradices* also known as *Rhizoglomus intaradices* basionym *Glomus intraradices*), the growth of *A. angustifolia* was favored, and autochthonous AMF were found to be more compatible with this species [10]. The application of AMF native to the Mezquital Valley in the state of Hidalgo also led to increases in the relative growth rate, stem biomass, root biomass, and water efficiency of *Agave salmiana* [11]. 

Inoculation with *Rhizophagus fasciculatus* (also known as *Rhizoglomus fasciculatus* basionym *Glomus fasciculatum*) or *R. intraradices* in *A. tequilana* improved daily net CO_2_ assimilation and photosynthesis and increased leaf thickness, although not growth [12]. Moreover, the inoculation of *A. tequilana* with *R. intraradices* and *Gluconoacetobacter diazotrophicus* or other diazotrophic bacteria was also observed to increase growth and the catalytic activity of the hydrolytic enzymes β-glucosidase, cellobiohydrolase, and en-do-1,4-β-D-glucanase in vitro [13]. In addition, micropropagated seedlings of *A. grijalvensis* inoculated with *R. fasciculatus* (basionym *G. fasciculatum*) and fertilized with P exhibited increases in the number of leaves, root length, and fructan concentrations in the stems and leaves [14]. 

The addition of different consortia of AMF (native to the rhizospheric soils of *Agave cupreata*) or commercial inoculum (Instituto Nacional de Investigaciones Forestales, Agrícolas y Pecuarias, INIFAP) to *A. inaequidens* has been reported to stimulated growth, although the degree was found to depend on the composition of each consortium [15]. It has also been found that inoculation with different autochthonous consortia of AMF improved the growth of *A. cupreata* [15]. However, when evaluating the potential of the AMF to protect against disease caused by *Fusarium oxysporum*, not all consortia were found to confer bioprotection [16]. 

Interestingly, the amount of crude protein and neutral detergent fiber was found to increase in *Agave americana* due to inoculation with the autochthonous *R. intraradices* from a commercial inoculum (INIFAP), although the amount of dry matter remained the same [17]. Similarly, beneficial and synergistic effects on the growth and production of sugars in *A. americana* were also observed due to inoculation with *R. fasciculatus* and *Penicillium* sp. as a phosphate solubilizer [18]. 

The paucity of information regarding the specific benefits produced by AMF in most *Agave* species must be addressed given the importance of this information for creating and implementing sustainable management, harvest, and conservation strategies. As such, the purpose of this study was to assess the effects of AMF (both allochthonous and autochthonous) on *A. maximiliana* by evaluating the growth variables and plant quality indicators associated with plant nutrition. We hypothesized that the effects produced by autochthonous AMF would be more beneficial than those produced by allochthonous AMF. We also compared the effects of both AMF types with those of chemical fertilization.

## 2. Results and Discussion

### 2.1. AMF Colonization

The percentage of mycorrhization increased over time (Figure 1), with the highest colonization percentage (64%) observed at 18 months after inoculation in all treatments (Table 1). The mycorrhizal colonization produced by the autochthonous consortium (Cn), individually evaluated allochthonous strains (Ce or Cc), and mixed allochthonous strains (Ce + Cc) was the same at each evaluation time (Table 1). 

These results are similar to those published by Robles-Martínez et al. [10] for *A. angustifolia* in which the inoculated autochthonous consortium of AMF resulted in the same levels of colonization as that of the allochthonous consortium (i.e., 27–37%) at 14 weeks, which is close to the 24-week evaluation period of this study. In contrast, the colonization percentages of *A. tequilana* plants (2 years and nine months in age) were greater than 50% [12] and 49% at 300 days [19], greater than 60% in *A. deserti* five months after inoculating 16-cm plants [8], ~80% in *A. salmiana* eight months after inoculation [11], and ~70% in the transformed roots of *A. salmiana* using *Agrobacterium rhizogenes* for the monoxenic cultivation of *R. intraradices* [20]. These results suggest that a direct relationship between plant age and the degree of AMF colonization is present in *Agave* species. Furthermore, it has been proposed that the high colonization values of older agave plants may be maintained by CAM metabolism, which improves the assimilation of CO_2_ assimilation and water retention thus allowing plants to allocate carbon compounds into AMF without jeopardizing their own requirements. Thus, maintaining AMF associations may not only be important for the growth of *A. angustifolia* but crucial for their survival [21].

The wide variation in the colonization percentages of the sampled roots was more evident at six months of age when considering the median values, although no differences were found between treatments (Table 1). In biological terms, changes in plant growth and development, including senescence of the radical system, as pointed out by Pimienta-Barrios et al. [12] for *A. tequilana*, help explain the most pronounced variation in mycorrhization levels at that age, although they were not statistically different. With regard to the intrinsic roles of AMF in generating this variation, it has been proposed that different species possess different strategies and colonization capabilities, with some species being better colonizers than others [22]. Plants also modulate colonization levels to varying degrees. Although the AMF interaction is non-specific, there appear to be patterns between plants and the AMF communities of their rhizospheres in tropical and humid [23], dry [24], and semi-dry [25] ecosystems, in which several *Agave* species are found. Variations among plant genomes also influence the degree of AMF colonization [26], which should be considered when evaluating the differences in AMF colonization of *A. maximiliana* in this study. This is particularly true in the short term given that the study plants were obtained from the seeds of wild plants. The high values of AMF colonization in this study suggest that a strong dependence exists between *A. maximiliana* and AMF, in which the energy expended by plants to maintain the mycorrhizal associations is justified by the benefits that are received. This would also likely apply to other *Agave* species. For example, it has been estimated that the dependence of *A. tequilana* on AMF is 62% [19]. However, this hypothesis requires further study.

### 2.2. AMF Spore Density in Inoculated Treatments

Spore quantities in the substrate varied among plant ages (Table 1). Nonetheless, the lowest quantities of spores were found at six months in all treatments. At 12 months, the highest spore quantities were found in two treatments containing allochthonous AMF species, namely Cc (915 spores in 100 g^−1^ of dry soil) and Ce + Cc (2922 spores in 100 g^−1^ of dry soil). At 18 months, treatment Cc showed the highest increase in the number of spores (4–16 times greater) when compared to those of the other treatments (Table 1). These results support the hypothesis that AMF species and plants exhibit two patterns of adaptation and sporulation. While some species are generalists and produce large numbers of spores while being associated with a wide variety of plants (allochthonous AMF), others are specialists and produce few spores [27]. In this regard, it has been proposed that specialist species belong to autochthonous AMF, which support mycorrhizal mycelium-mediated interactions, and that the distinct species that influence these associations do not require extensive sporulation. This hypothesis has been supported in semi-woody plants with long life cycles [28]. Agave plants are long-lived, which may allow for this hypothesis to be supported in these species as well. 

### 2.3. Plant Growth, Nutrition, and Metabolism

Both the fresh weight and height of the plants steadily increased over the study. From the sixth month onward, the plants in the treatments inoculated with AMF (Cn, Ce, Cc, and Ce + Cc) showed greater fresh weight gains and weighed 10 times more than those of the control treatments (Tf and Tn) at 18 months (Figure 2). It should be noted that an antagonistic effect was observed when the Ce and Cc inoculums were mixed (Ce + Cc). Similarly, a positive effect was observed with regard to the heights of plants treated with AMF (Figure 3). At the end of the experiment, AMF-treated plants were three to four times larger than the control plants (Figure 4). The growth curves (weight and height) of the AMF-treated plants showed two periods of active growth, the first beginning at 6 months and the second beginning at 14 months. 

At the earliest age (i.e., 6-month-old plants), the best responses in terms of plant development were evident in the increases in the base diameter, number of leaves, and leaf length. However, the plants in the Ce + Cc treatment showed similar leaf lengths to those of both control treatments. The highest radical length and aerial biomass values were observed in the Cn and Ce treatments, while the highest and lowest number of leaves were found in the Cc and Ce + Cc treatments, respectively (Table 2). In 12-month-old plants, the effects of the AMF were clear in all developmental variables with the exception of root length in the Cc treatment, which was clearly different from those of all other treatments, and fresh aerial biomass, which was the same as those of the Ce + Cc treatment and both controls (Table 2). The most notable results that reflected the beneficial effects of the interaction between AMF and agave plants were observed at 18 months. At this time, all variables showed increases in AMF-treated plants, with the exception of the root length in the Ce treatment, which was the same as that of both controls (Table 2). 

In a previous study with *A. tequilana*, no effects of mycorrhization were found on the quantity or length of the leaves [12]. However, Montoya-Martínez et al. [19] found higher values for height, biomass, leaf area, and number of leaves in the same species, although with only one autochthonous AMF consortium of *A. cupreata*. In contrast, inoculation with *R. fasciculatus* (basionym *G. fasciculatum*) increased the fresh weight, number of leaves, and root length in *A. grijalvensis* [14]. A growth pattern similar to that found with height in *A. maximiliana* was found by Montoya-Martínez et al. [19] in *A. tequilana*, who observed a steady increase. However, at the end of their experiment, they found that only one of the AMF consortia resulted in a greater number of leaves compared to those of the control plants.

The results obtained with the mixture of the allochthonous inoculums (Ce + Cc) show that the interaction between both species was not additive as would have been expected when adding together the benefits that were observed for each individually. However, the effects on the plants in this treatment indicate the existence of interspecific competition between both fungi. In in vivo studies with *Glomus tenue*, *Glomus fasciculatum*, and *Gigaspora decipiens* and in vitro studies, HMA species have been found to reciprocally limit extrarradical and intrarradical mycelial growth by strongly competing with each other for plant resources [29,30]. This phenomenon directly affects nutrient transfer to the plant and consequently limits plant growth. With regard to species that are more competitive, Engelmore et al. [30] proposed that those that are more closely linked phylogenetically may be more competitive with each other, as was observed with species used as allochthonous inocula that belong to the same genus. However, Cano and Bago [31] noted that a similar phenomenon is apparent in unrelated genera such as *Glomus sp.* and *Gigaspora sp*. Nonetheless, little is known regarding competition between HMA, which makes conducting research aimed at elucidating the competition and antagonism present between HMA species both important and interesting.

Nutrition was evaluated by the content of different minerals at 18 months. When compared to the values of the control plants, the AMF-treated plants exhibited higher levels of P (four to six times higher), K, and Mg (about twice as high) but low levels of Fe. The content of Ca in AMF-treated plants was similar to that of the fertilized control plants, and N, B, Cu, Mn, and Zn content was the same among the AMF-treated and control plants (Table 3). 

Robles-Martínez et al. [10] found no differences in N content between AMF-treated and control *A. angustifolia* plants, although P content improved with AMF treatment. However, the increase in P content only occurred with four of the different autochthonous AMF consortia and with allochthonous *R. intraradices* (basionym *G. intraradices*). On the other hand, García-Martínez et al. [32] found that *A. potatorum* Zucc and a group of *Agave* spp. (agave coyote) showed improved growth with AMF inoculation and P application (but not due to the interaction of these two substances). These authors, along with Quiñones-Aguilar et al. [15] (*A. inaequidens*), Montoya-Martínez et al. [19] (*A. tequilana*), and Trinidad-Cruz et al. [16] (*A. cupreata*), who inoculated their study plants with a multispecies consortium of *A. cupreata*, concluded that the acquisition of P by AMF-treated plants is independent of its concentration in the soil or substrate given that the capacity and efficiency of AMF to absorb P varies among species (interspecific variation), among strains of the same species (intraspecific variation), and within the same host plant species [16]. 

The results of this study with *A. maximiliana* indicate that the increase in P, K, and Mg concentrations was noticeably greater in AMF-treated plants regardless of the origin of the inoculum (i.e., allochthonous or autochthonous). With respect to the role of K and Mg, Ochoa-Meza et al. [21] found that the soil concentrations of these elements have an important influence on mycorrhizal associations in *A. angustifolia*. However, Ríos-Ramírez et al. [33] found that the increases in the concentrations of K, Mg, P, Ca, and Fe in AMF-free plants were directly related to the amount of nutrients they received via irrigation. Under field conditions, *A. angustifolia* and *A. karwinski* both show high concentrations of Mg, Ca, and Fe [34]. These results highlight the importance of these elements for *A. maximiliana*, *A. angustifolia*, and *A. karwinski*.

As with *A. tequilana* [35], there are no reference values to determine if the low content of Fe in AMF-treated *A. maximiliana* was insufficient, although it was evident that these plants did not show signs of nutritional deficiencies due to a lack of N or any microelements as a result of inoculation. In contrast, the control plants, especially those that were not fertilized nor inoculated with AMF (Tn), showed clear signs of nutrients deficiencies that were evident not only by the observed low plant growth (size, fresh biomass, root area, root length, number of leaves, and leaf length) and the presence of chlorosis symptoms. Deficiencies of N, K, P, Ca, Mg, and Fe in *A. tequilana* plants under conditions of nutritional restriction have been found to result in small plants and leaves and chlorosis [36].

Studies that have evaluated fertilization in the absence of AMF have shown the importance of N for plant growth and productivity, e.g., *A. lechuguilla* [37], *A. angustifolia* [21], *A. tequilana* [38], and *A. potatorum* in 5-year plants [39]. This information may partially explain the lack of differences observed in N concentrations between AMF-treated and untreated *A. maximiliana* plants. The limited amount of information regarding the roles of AMF in N acquisition by *A. maximiliana* and other *Agave* species has led to the development of multiple hypotheses that must be evaluated. One hypothesis posits that *Agave* species are relatively unresponsive to variations in N content in their first years of development, although their responsiveness increases as the plant matures. Another hypothesis suggests that *Agave* species are dependent on N to varying degrees based on their needs. Finally, another hypothesis proposes that AMF do not contribute to N uptake, which is completely contrary to what has been observed with P in this and other studies.

Total reducing sugars (TRS) and free sugars (FRS) were evaluated as indicators of plant metabolism. The content of TRS was higher in the control plants than in the AMF-treated plants, while the content of FRS was higher in the plants of the Cn and Cc + Ce treatments (Table 3). In *A. angustifolia* and *A. karwinski*, TRS values of 21.16 and 27.29%, respectively, have been reported in the piñas [34], while a TRS value of 34% has been reported in AMF-free *A. cocui* [40]. The low concentrations of reducing sugars in AMF-treated plants suggest that they may have been metabolized. This phenomenon has been confirmed with *A. grijalvensis*, in which active enzymatic involvement was identified that was responsible for sugar hydrolysis along with the increased production of fructooligosaccharides, glucose, and fructose in seedlings treated with *R. fasciculatum* [15]. Although very little information is available on this subject, based on the criteria for *A. tequilana* [41], the TRS content in AMF-treated *A. maximiliana* indicates that these plants were in good conditions given that the TRS values were greater than 25%, with the exception of those in the Ce treatment (22%).

## 3. Materials and Methods

### 3.1. Plant Material

*Agave maximiliana* plants were grown from seeds collected from a wild population in the *Quercus* forest of Mascota, Jalisco, which is located in the western coast-sierra region (20°23.115′ N, 104°45.575′ W, 1254 m a.s.l.). The seeds were manually cleaned and selected using the flotation method. The selected seeds were superficially disinfected with 0.3% sodium hypochlorite for 15 min, rinsed twice (5 min) with water, and soaked in sterile distilled water for 12 h. Once disinfected, the seeds were placed in polystyrene germinators with 30 mL of a sterile vermiculite:peat mixture (8:1 *v*/*v*; pH 6 ± 0.2) [42]. Germinators were irrigated at field capacity with sterile distilled water and packed with adhesive plastic to maintain moisture. One-month-old seedlings (3–4 cm long) were transplanted into containers with 5 kg of sandy loamy soil (0.28% organic matter and 2.18 mg kg^−1^ P).

### 3.2. AMF Inoculums

Strains of *C. claroideum* and *C. etunicatum* (i.e., the allochthonous AMF) were obtained from the AMF collection of the Centro de Investigación en Ciencias Biológicas of the Universidad Autónoma de Tlaxcala. The strains were isolated from hardened volcanic soil (brown tepetate) from the municipality of Hueyotlipan, Tlaxcala. The consortium of autochthonous AMF was obtained from propagation pots in which rhizospheric soil of *A. maximiliana* was mixed with sterile sand (ratio 1:1, *v*/*v*) and trap plants consisting of corn (*Zea mays* L.), runner bean (*Phaseolus coccineus* L.), and coriander (*Coriandrum sativum* L.). These pots were kept in a greenhouse and irrigated at field capacity for six months. Three weeks before the end of this period, irrigation was interrupted to promote fungi sporulation.

### 3.3. Treatments

Six treatments were established: (1) consortium of autochthonous AMF of the *A. maximiliana* rhizosphere (Cn), (2) *C. etunicatum* (Ce), (3) *C. claroideum* (Cc), (4) *C. etunicatum* + *C. claroideum* (Ce + Cc), (5) control with fertilization and without mycorrhizal inoculation (Tf), and (6) control without fertilization or mycorrhizal inoculation (Tn). Each treatment included 18 plants that were distributed in a random block design within a greenhouse where they were kept for 18 months. Fertilization in the Tf treatment was made up of nutrients (180-80-60 mg Kg^−1^ of N-P-K) administered in a solution prepared with KNO_3_ (1 M), Ca(NO_3_)_2_ (1 M), and KH_2_PO_4_ (0.01 M) [43]. The nutrient solution was administered monthly. The P concentration was neither restrictive nor toxic to the plants. The pH of the substrate was 7 ± 0.2 and did not have negative effects on the nutrients, as it behaved like an inert substrate.

### 3.4. AMF Inoculation and Experiment

Each AMF-treated plant was inoculated with 100 spores when they were transplanted [44]. The AMF species in the Cn treatment were *Acaulospora laevis*, *A. morrowiae*, *A. spinosa*, *C. claroideum*, *C. etunicatum*, *Funneliformis geosporus*, and *Glomus microaggregatum*. In the Ce + Cc treatment, the plants were inoculated with 50 spores of each species. The taxonomic identification of the spores produced by the autochthonous fungi and known strains was performed with intact spores in good condition. The morphological characteristics of these spores were observed with and without Melzer reagent using Nomarski interference contrast microscopy (Nikon Optiphot-2; Nikon, Tokyo, Japan) and compared with the descriptions available for AMF species from the literature [45,46,47,48,49,50,51,52] and those available in the AMF phylogeny website (http://www.amf-phylogeny.com). The plants were watered every third day for the first three months and every four days for the remaining 18 months of the experiment.

### 3.5. Quantification of AMF Colonization

The colonization percentage was evaluated in the roots of three plants per treatment at 6, 12, and 18 months. The roots were washed with running water and stained with Trypan blue [53] to visualize the fungal structures (cenocytic hyphae, vesicles, hyphal coils, arbuscles, or spores). Approximately 20 segments (2-cm in length) of the dyed roots of each plant were placed on slides and observed with a light field microscope at 20× and 40× (Nikon Optiphot-2) according to the method described by McGonigle et al. [54].

Sampling times were defined based on agave life cycles and the results of previous experiments with other *Agave* species [12,17], which have shown little or no short-term effects. We also wanted to favor the establishment and functions of inoculated AMF, as the autochthonous inoculum contained species of *Acaulospora* spp. The spores of this species have a latency period that can last three months [55,56] (Gazey et al., 1993; Giovanetti, 2000).

### 3.6. Recovery and Estimation of AMF Spore Density

Samples of the rhizosphere soil from each plant (50 g) were wet sieved, decanted, and centrifugated in a sucrose gradient (20% and 60%) [57]. The extracted spores were fixed with polyvinyl alcohol-lactic acid-glycerol and Melzer reagent in a ratio of 1:1 (*v*/*v*).

### 3.7. Determination of Physiological Variables

At 6, 12, and 18 months after being transplanted, the development parameters of the plants (e.g., leaf number, longest leaf length, base diameter, fresh biomass of the aerial and root portions, and longest root length) were determined.

### 3.8. Reducing Sugars 

At the end of the 18-month experiment, the content of free and total reducing sugars in the middle section of the leaves was determined using the dinitrosalicyl acid (DNS) method with a microplate [58]. The agave leaves were crushed in a food processor, and the sample was centrifuged at 12,000× *g* for 20 min. Then, a 1 mL aliquot was hydrolyzed with phosphoric acid 1M (pH 2) for 30 min at 70 °C. The quantification of free-reducing sugars content was also performed with the supernatant, although without hydrolyzing. A total of 20 µL of the sample was placed in a 96-well microplate and mixed with 200 µL of the DNS reagent (0.1% 3,5-dinitrosalicyl acid, *w*/*v*) and sodium potassium tartrate (30%, *w*/*v*) in NaOH (0.4 M). This procedure was performed for all samples and a standard. The microplate was sealed and placed in the microwave at maximum power for 4 min and then cooled to room temperature. Absorbance was read at 570 nm. Sugar content was determined using a standard fructose curve of 0 to 15 mg mL^−1^.

### 3.9. Determination of Foliar Nutrients

The micro and macroelements in plant tissue were also determined. For this, leaves from each treatment were placed in an oven at 60 °C for 5 days. After which, the samples were ground, and total nitrogen content was determined by the micro-Kjeldahl method with volumetric titration [59]. The content of Ca, K, Mg, Cu, Fe, Mn, and Zn was determined from a calcium-cinate sample at 475 °C by atomic absorption spectrophotometry, and the content of P and B were determined by the colorimetric method described by [60].

### 3.10. Statistical Analysis

The colonization percentages were evaluated with contingency tables and X^2^ tests (general and treatment pairs; α = 0.05). Spore and leaf quantities were analyzed by a Kruskall–Wallis test followed by a Mann–Whitney U test (α = 0.05) [61]. The variables of base diameter; leaf length; root length; aerial biomass; radical biomass; free and total reducing sugar content and the content of P, B, Fe, Mn, Zn, K and N were evaluated using a simple analysis of variance (ANOVA) followed by a means test (α = 0.05). The N, K, Ca, and Mg data were arcsine transformed prior to being analyzed. All data were processed with JMP v. 4.0.2 [62].

## 4. Conclusions

The results of this study highlight notable benefits with regard to the nutrition, growth, and metabolism of *A. maximiliana* due to mycorrhization, which have also been observed in other *Agave* species. In light of this, it is clear that AMF should be considered key biological elements for the management of *A. maximiliana*, as they notably promote the production of robust plants of good quality. In doing so, *A. maximiliana* may be reintroduced into agroforestry systems and plants may be produced that are suitable for cultivation systems. All of these actions would support and promote the conservation of this phytogenetic resource.

*Agave maximiliana* plants are susceptible to colonization by autochthonous and allochthonous AMF, and colonization increases over time. The abundant sporulation of the allochthonous AMF and low spore production of autochthonous AMF indicate the adaptation of the latter to agave plants. In terms of nutrition, *A. maximiliana* substantially benefited from mycorrhization regardless of the origin of the AMF, which promoted the acquisition of P, K, Mg, and Fe but not of N, B, Cu, Mn, or Zn. The autochthonous consortium and individual allochthonous AMF produced similar responses among the different growth variables, although the mixture of the two allochthonous AMFs produced responses that were not as favorable. The high content of total reducing sugars in the plants of the control treatments and in fertilized, not fertilized, or AMF-treated plants as well as the macro and micronutrient content that benefited from the AMF inoculation, indicate that these sugars were actively utilized and metabolized and contributed to better growth in AMF-treated plants. Mycorrhization produces notable growth in *A. maximiliana* and results in vigorous radical systems that make plants suitable for transplant in shorter time periods than those of conventional production systems, which will support important management and conservation strategies.

Until 2007, *A. maximiliana* was considered abundant in Jalisco [63]. Since then, wild *A. maximiliana* populations have critically declined due to overexploitation as a consequence of raicilla production, which has put this species at risk of being eradicated from agroforestry systems [64]. To conserve this species without affecting raicilla production, management practices must be improved, its persistence in agroforestry systems must be promoted, and monoculture systems under agroecological schemes must be modified [64,65].

## Figures and Tables

**Figure 1 plants-12-00535-f001:**
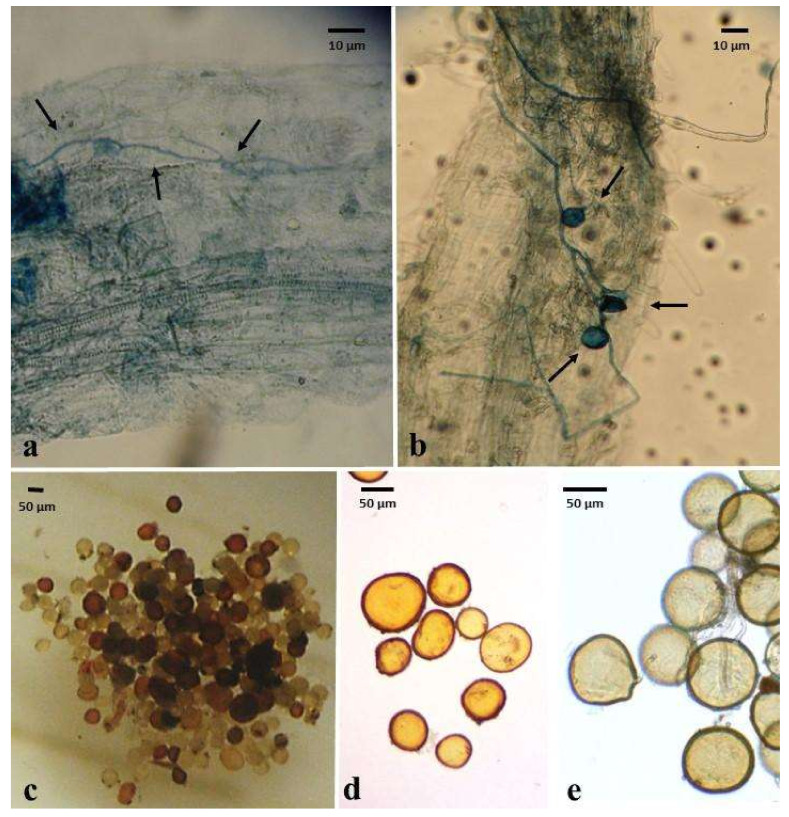
Hyphae, vesicles, and spores of arbuscular mycorrhizal fungi (AMF). Abbreviations: consortium of autochthonous AMF of the *Agave maximiliana* rhizosphere (Cn), *Claroideoglomus etunicatum* (Ce), and *Claroideoglomus claroideum* (Cc) (**a**) AMF hyphae in the roots of *A. maximiliana* roots after six months of treatment (Cc). (**b**) AMF Hyphae and vesicles in *A. maximiliana* roots after 12 months of treatment (Cn). (**c**) Spores of Cn. (**d**) Spores of Ce. (**e**) Spores of Cc.

**Figure 2 plants-12-00535-f002:**
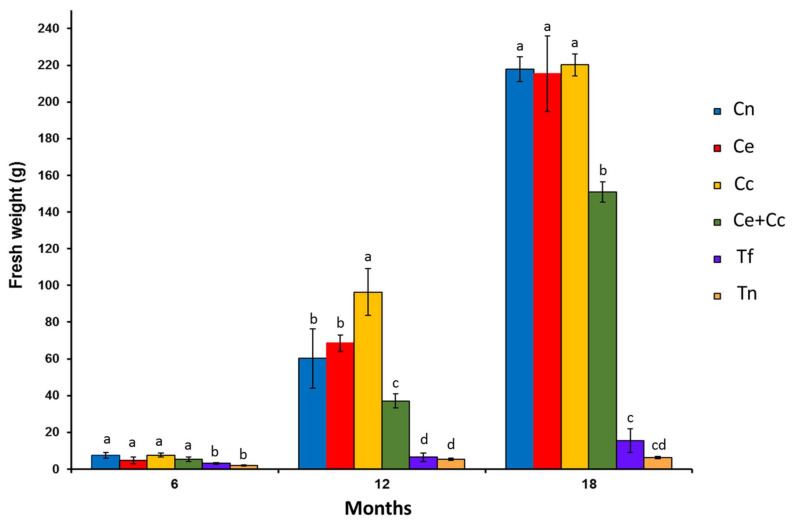
Fresh plant weight of *Agave maximiliana* at 6, 12, and 18 months after inoculation with arbuscular mycorrhizal fungi (AMF). Treatments: (1) consortium of autochthonous AMF of the *A. maximiliana* rhizosphere (Cn), (2) *Claroideoglomus etunicatum* (Ce), (3) *Claroideoglomus claroideum* (Cc), (4) *C. etunicatum* + *C. claroideum* (Ce + Cc), (5) control with fertilization and without mycorrhizal inoculation (Tf), and (6) control without fertilization or mycorrhizal inoculation (Tn). Age 6 months, n = 18; age 12 months, n = 13; age 18 months, n = 8. The bars represent the means ± standard error. An analysis of variance was followed by a means test (*p* ≤ 0.05). Different letters (by age) indicate significant differences between treatments.

**Figure 3 plants-12-00535-f003:**
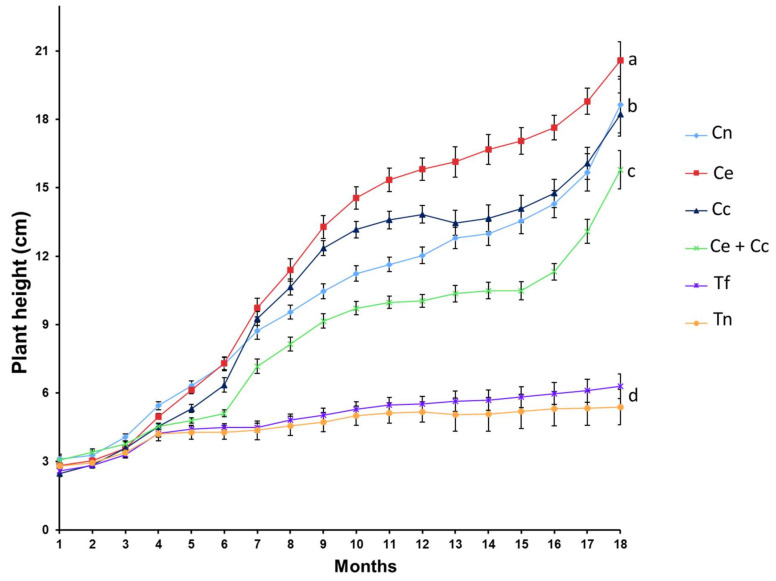
Heights of *Agave maximiliana* plants over 18 months after being inoculated with arbuscular mycorrhizal fungi (AMF). Treatments: (1) consortium of autochthonous AMF of the *A. maximiliana* rhizosphere (Cn), (2) *Claroideoglomus etunicatum* (Ce), (3) *Claroideoglomus claroideum* (Cc), (4) *C. etunicatum* + *C. claroideum* (Ce + Cc), (5) control with fertilization and without mycorrhizal inoculation (Tf), and (6) control without fertilization or mycorrhizal inoculation (Tn). Ages 1–6 months, n = 18; ages 7–12 months, n = 13; ages 13–18 months, n = 8. Data are means ± standard error. Different letters (at 18 months) indicate significant differences between treatments.

**Figure 4 plants-12-00535-f004:**
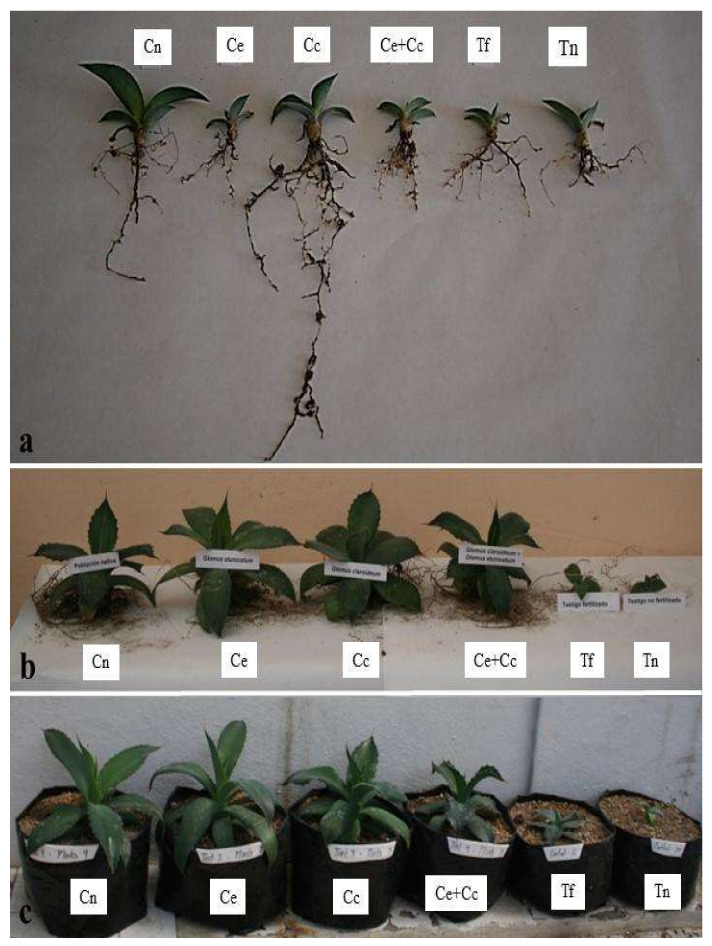
*Agave maximiliana* plants of four treatments that included inoculation with arbuscular mycorrhizal fungi (AMF) and two control treatments. Treatments: (1) consortium of autochthonous AMF of the *A. maximiliana* rhizosphere (Cn), (2) *Claroideoglomus etunicatum* (Ce), (3) *Claroideoglomus claroideum* (Cc), (4) *C. etunicatum* + *C. claroideum* (Ce + Cc), (5) control with fertilization and without mycorrhizal inoculation (Tf), and (6) control without fertilization or mycorrhizal inoculation (Tn). (**a**) Plants at 6 months of treatment. (**b**) Plants at 12 months of treatment. (**c**) Plants at 18 months of treatment.

**Table 1 plants-12-00535-t001:** Mycorrhizal colonization (%) and the number of spores of arbuscular mycorrhizal fungi (AMF) per 100 g dry soil. Colonization of *Agave maximiliana* by the consortium of autochthonous AMF of (1) the *A. maximiliana* rhizosphere (Cn) and the allochthonous AMF of (2) *Claroideoglomus etunicatum* (Ce), (3) *Claroideoglomus claroideum* (Cc), and (4) *C. etunicatum* + *C. claroideum* (Ce + Cc) at 6, 12, and 18 months under greenhouse conditions.

		Mycorrhizal Colonization (%) *		Number of Spores/100 g of Soil **
Age (Months)	Treatment	Median	Minimum and Maximum Values of Quartiles	
6	Cn	23.60 a	10.55–36.66	20.13 ± 2.82 a
	Ce	6.44 a	1.68–11.2	32.85 ± 5.95 a
	Cc	22.21 a	11.66–32.77	16.22 ± 3.05 a
	Ce + Cc	31.85 a	29.55–34.16	26.67 ± 10.47 a
12	Cn	19.615 a	18.33–20.90	17.07 ± 10.50 b
	Ce	38.74 a	35–42.48	37.26 ± 11.00 b
	Cc	37.57 a	32.22–42.93	2922.68 ± 1600.50 a
	Ce + Cc	22.05 a	12.50–31.60	915.46 ± 314.00 a
18	Cn	55.27 a	47.77–62.77	51.05 ± 8.66 b
	Ce	48.33 a	43.33–53.33	188.16 ± 155.22 b
	Cc	63.88 a	62.77–65.00	831.27 ± 132.72 a
	Ce + Cc	64.44 a	61.66–67.22	87.81 ± 34.67 b

* n = 3, *X^2^* general and by pairs, α = 0.05, f.d. 3, *v* = 4; ** n = 5, media ± standard error, Kruskall-Wallis followed by Mann-Whitney U test, α ≤ 0.05. Different letters in each column (by age) represent significant differences between treatments.

**Table 2 plants-12-00535-t002:** Growth indicators of *Agave maximiliana* inoculated with arbuscular mycorrhizal fungi (AMF) at 6, 12, and 18 months under greenhouse conditions. Treatments: (1) consortium of autochthonous AMF of the *A. maximiliana* rhizosphere (Cn), (2) *Claroideoglomus etunicatum* (Ce), (3) *Claroideoglomus claroideum* (Cc), (4) *C. etunicatum* + *C. claroideum* (Ce + Cc), (5) control with fertilization and without mycorrhizal inoculation (Tf), and (6) control without fertilization or mycorrhizal inoculation (Tn).

Age (Months)	Treatment	Growth Variables				
		Base diameter (mm)	Number of leaves ^+^	Foliar length (cm)	Radicle length (cm)	Radicle biomass (g)
6 *	Cn	9.97 ± 0.13 a	5.55 ± 0.14 ab	7.25 ± 0.30 a	19.4 ± 4.58 a	0.39 ± 0.15 a
	Ce	10.11 ± 0.06 a	5.55 ± 0.66 ab	7.28 ± 0.28 a	10.52 ± 1.69 b	0.15 ± 0.04 a
	Cc	10.06 ± 0.14 a	6.27 ± 0.21 a	6.33 ± 0.32 a	12.42 ± 2.55 ab	0.28 ± 0.07 a
	Ce + Cc	9.63 ± 0.18 a	5.16 ± 0.14 b	5.01 ± 0.15 b	9.7 ± 2.06 ab	0.26 ± 0.07 a
	Tf	8.40 ± 0.28 b	4.22 ± 0.19 c	4.48 ± 0.16 b	8.6 ± 0.41 b	0.17 ± 0.03 a
	Tn	8.16 ± 0.52 b	4.00 ± 0.28 c	4.27 ± 0.30 b	7.12 ± 0.44 b	0.14 ± 0.02 a
12 **	Cn	30.05 ± 0.84 ab	10.46 ± 0.29 a	12.03 ± 0.36 c	21.64 ± 4.94 b	2.48 ± 0.78 a
	Ce	27.27 ± 0.60 bc	10.61 ± 0.18 a	15.82 ± 0.49 a	23.4 ± 2.15 ab	2.67 ± 0.37 a
	Cc	31.94 ± 0.84 a	11.15 ± 0.27 a	13.83 ± 0.39 b	41.9 ± 6.87 a	3.88 ± 0.35 a
	Ce + Cc	24.70 ± 0.62 c	8.69 ± 0.20 b	10.03 ± 0.28 d	32.52 ± 5.73 ab	2.20 ± 0.23 a
	Tf	12.44 ± 0.77 d	5.76 ± 0.25 c	5.51 ± 0.33 e	17.34 ± 3.67 b	0.30 ± 0.11 b
	Tn	11.01 ± 0.98 d	4.84 ± 0.41 c	5.16 ± 0.44 e	15.7 ± 2.07 b	0.32 ± 0.07 b
18 ***	Cn	46.20 ± 0.69 a	16.37 ± 0.70 ab	18.65 ± 1.25 ab	46.6 ± 3.72 a	7.72 ± 0.84 a
	Ce	44.67 ± 1.52 ab	16.12 ± 0.39 ab	20.60 ± 0.82 a	34.2 ± 4.70 b	7.64 ± 1.03 a
	Cc	48.48 ± 1.48 a	17.25 ± 0.61 a	18.22 ± 0.95 ab	38.4 ± 2.65 ab	10.80 ± 1.59 a
	Ce + Cc	38.91 ± 1.39 b	14.12 ± 0.66 b	15.78 ± 0.84 b	39.6 ± 1.53 ab	8.00 ± 0.98 a
	Tf	16.12 ± 1.63 c	7.5 ± 0.42 c	6.27 ± 0.54 c	25 ± 4.81 bc	0.56 ± 0.21 b
	Tn	12.26 ± 1.84 c	6.12 ± 0.89 c	5.37 ± 0.77 c	18.6 ± 1.80 c	0.28 ± 0.05 b

* n = 18, ** n = 13, *** n = 8, mean ± standard error, analysis of variance followed by means test, *p* ≤ 0.05; + Kruskall–Wallis followed by Mann–Whitney U test, α ≤ 0.05. Different letters in each column (by age) indicate significant differences between treatments.

**Table 3 plants-12-00535-t003:** Content of reducing sugars and mineral nutrients in *Agave maximiliana* 18 months after inoculation with autochthonous arbuscular mycorrhizal fungi (AMF). Treatments: (1) consortium of autochthonous AMF of the *A. maximiliana* rhizosphere (Cn), (2) *Claroideoglomus etunicatum* (Ce), (3) *Claroideoglomus claroideum* (Cc), (4) *C. etunicatum* + *C. claroideum* (Ce + Cc), (5) control with fertilization and without mycorrhizal inoculation (Tf), and (6) control without fertilization or mycorrhizal inoculation (Tn).

Treatment	Reducing Sugars		Minerals									
	Free (μg/mL)	Total (μg/mL)	N (%)	K (%)	Ca (%)	Mg (%)	P (ppm)	Bo (ppm)	Cu (ppm)	Fe (ppm)	Mn (ppm)	Zn (ppm)
Cn	230 ± 0.04 a	310 ± 0.14 b	1.44 ± 0.05 a	4.25 ± 0.12 a	2.44 ± 0.14 a	1.31 ± 0.07 a	2785.58 ± 488.46 b	40.41 ± 1.95 a	<4.81	125.44 ± 6.90 b	43.93 ± 3.36 b	15.61 ± 3.21 b
Ce	160 ± 0.02 b	220 ± 0.01 b	1.54 ± 0.08 a	4.09 ± 0.16 a	2.27 ± 0.08 a	1.41 ± 0.02 a	2391.85 ± 165.34 b	41.56 ± 3.05 a	<4.81	160.44 ± 27.01 b	61.91 ± 3.41 a	17.34 ± 0.48 b
Cc	180 ± 0.04 b	260 ± 0.03 b	1.77 ± 0.07 a	4.39 ± 0.05 a	2.24 ± 0.04 a	1.37 ± 0.03 a	3350.93 ± 203.47 a	44.64 ± 0.83 a	<4.81	163.26 ± 59.51 b	47.86 ± 6.38 b	21.75 ± 1.96 a
Ce + Cc	220 ± 0.06 a	350 ± 0.07 b	1.48 ± 0.06 a	4.95 ± 0.28 a	2.38 ± 0.27 a	1.58 ± 0.04 a	3210.83 ± 356.44 a	43.74 ± 1.55 a	<4.81	137.79 ± 30.06 b	33.20 ± 2.72 c	23.94 ± 1.07 a
Tf	140 ± 0.06 c	780 ± 0.14 a	1.49 ± 0.39 a	2.77 ± 0.31 b	2.15 ± 0.17 a	0.99 ± 0.08 b	620.29 ± 81.72 c	43.36 ± 1.04 a	<4.81	197.24 ± 495.85 a	34.76 ± 5.79 c	16.47 ± 8.40 b
Tn	120 ± 0.02 c	620 ± 0.18 a	1.47 ± 0.53 a	1.93 ± 0.10 b	1.67 ± 0.06 b	0.73 ± 0.06 b	401.70 ± 26.72 c	43.97 ± 3.38 a	<4.81	169.22 ± 98.88 a	28.76 ± 8.33 c	9.13 ± 0.86 c

n = 5, mean ± standard error; analysis of variance followed by means test, α = 0.05. Different letters in each column indicate significant differences between treatments.

## Data Availability

Not applicable.

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
