# Peer review of "Physiological Responses of Agave maximiliana to Inoculation with Autochthonous and Allochthonous Arbuscular Mycorrhizal Fungi"

_plants, 2023, doi:10.3390/plants12030535_

Round 1

Reviewer 1 Report

This study aimed to evaluated the effects of autochthonous and allochthonous AMF on growth and nutrient assimilation in Agave maximiliana. In this study, there are 6 treatments and data of mycorrhizal colonization, plant height, fresh weight, growth indicators, reducing sugars and mineral nutrients. The results shows that AMF improve the growth and nutrient assimilation of Agave maximiliana, and autochthonous AMF is more suitable for agave plants. But previous studies have reported autochthonous AMF are more compatible with the agave, just different species. Furthermore, some interesting results need more discussion. For example, the results about antagonistic effect between two kinds of allochthonous AMF and the poor performance of Tf treatment. Maybe you can add some results about production of raicilla. It will be more helpful to take actions for conservation of Agave maximiliana. Besides there is a little bit unclear about numbers of sample. It seems not uniform among different indicators. Some other issues are listed below.

 Other comments:

Line 157 I doubt whether this expression of “AMF ratios” is accurate.

 Line 170 Please check there should be “Fig. 1” or “Table 1”.

 Line 278-279 Do you mean acquisition of P by the whole plants? Is it the mycorrhizal pathway plus the root pathway? Even though the acquisition of P by plants will be affected by P concentration in the soil or substrate.

Reviewer 2 Report

The current study reports the beneficial effects of Arbuscular mycorrhizal fungi (AMF) on the growth and development of A. maximiliana. The study reports the positive effect of allochthonous and autochthonous AMF on the A. maximiliana, which is evident in all growth indicators, sugar, and mineral (P, K, Ca, Mg, and Fe) content. AMF significantly improved all growth parameters of A. maximiliana regardless of the origin of the inoculums. While similar studies have been conducted in other species of Agave, this is the first ever report in A. maximiliana.  Overall, the study endorses the use of AMF species as key biological elements for the management of A. maximiliana.

MAJOR POINTS:

·         The idea of including controls with (Tf) and without fertilisation (Tn) (and without AMF inoculation) is commendable. It is important for studies pertaining AMF to compare the performance of plants with the ones that are subjected to fertilisers so as to evaluate the efficiency of AMF in improving the plant performance in ‘realistic’ terms.

However, the authors should give a brief information about what kind of fertilisers were used in the study. Were they organic or synthetic and what was the nutrient composition (given that the authors have estimated nutrient concentration of all the plants, it becomes all the more important to have a note of the mineral composition of the fertilisers)?

Also, in what concentration were the Tf plants given the fertilisers? Was it comparable/ equivalent in any respect to the amount of inocula given per AMF treatment? Because, surprisingly, the plants subjected to fertilisation have underperformed w.r.t growth parameters over progressing growth stages. More so, they are comparable to the ones that were not given any kind of external fertiliser treatment (Tn) (Fig 2)

·         As observed by the authors, an antagonistic effect on the growth (fresh weight and height) was recorded when the Ce and Cc inoculums were mixed (Ce+Cc). What could be the possible reasons of that??

·         It is appreciated that the authors have disintegrated the analysis of growth parameters as an effect of AMF on A. maximiliana into three developmental stages, i.e., 6 - 12 - and 18 months after treatment. What is the relevance of choosing these developmental stages for studying the (favourable) influence of AMF on the host plant?

·         L 386: Correct the spellings of the dye ‘Trypan blue’

·         What were the reasons of isolating the allochthonous AMF strains from hardened ‘volcanic’ soil?

·         How did the authors confirm that the plants were actually colonised by that very strain of AM fungus with which they inoculated them.

·         In the ‘material and method’ section, discuss the reducing sugar and foliar nutrients under separate sub heads.

·         At what growth stage were reducing sugars (free and total) quantified?

·         Replace 3.1 head with ‘Biological/Plant Material’

·         L 402-403: What solvent/extraction buffer did you use to obtain the extract?

·         L 159- 161: “Variations among plant genomes also influence the degree of AMF colonization which should be considered when evaluating the differences in AMF colonization of A. maximiliana in this study”. How is the influence of plant genome variation on colonisation percentage applicable to this study where only one genotype of the host plant is used? In this study, it is rather a matter of the variability in AMF strains that are being used.

·         The standard abbreviation of grams is g not gs. Please correct the same in the manuscript.

·         As established in several studies, extremely high concentration of P in soil sabotages the growth and proliferation of AMF, thereby affecting colonisation and hence the acquisition of (absorbable) P by AMF for the host plant. In the light of this fact, can the authors comment on (L 278-279) “acquisition of P by AMF-treated plants is independent of its concentration in the soil or substrate”

·         Relevance of section 2.4 “AMF and the management and conservation of A. maximiliana” is not understood under the ‘Result and Discussion’ head. Either the section can be clubbed with the conclusion section or a part of it can be added to the introduction section

·         L 431-432: The abundant sporulation of the allochthonous AMF and low spore production of autochthonous AMF indicate the suitability of the latter to agave plants. Clarify?

·         L 441: What ‘transformation’ process are authors referring to?

·         There is scope of improvement in the discussion section, especially that of reducing sugars. The major parameters assessed should be discussed in correlation with each other rather than discussing them separately. Because everything is ultimately resulting in improved growth and robust performance of the plant. Moreover, English language can also be improved throughout the manuscript.

Round 2

Reviewer 2 Report

The authors have addressed the comments and incorporated the suggested changes.